# A Narrative Review of Compassion Focused Therapy on Positive Mental Health Outcomes

**DOI:** 10.3390/bs14080643

**Published:** 2024-07-25

**Authors:** Yasuhiro Kotera, Julie Beaumont, Ann-Marie Edwards, Matthew Cotterill, Ann Kirkman, Aiesha Carew Tofani, Merly McPhilbin, Simran Takhi, Kristian Barnes, Olamide Todowede, Benjamin-Rose Ingall, Kenichi Asano, Kohki Arimitsu

**Affiliations:** 1Faculty of Medicine and Health Sciences, University of Nottingham, Nottingham NG7 2RD, UKolamide.todowede@nottingham.ac.uk (O.T.);; 2Center for Infectious Disease Education and Research, Osaka University, Suita 565-0871, Japan; 3School of Psychology, Queen’s University Belfast, Belfast BT7 1NN, UK; jbeaumont01@qub.ac.uk; 4University of Essex Online: Kaplan Open Learning Essex Ltd., Wivenhoe Park, Colchester CO4 3SQ, UK; 5College of Health, Psychology and Social Care, University of Derby, Derby DE22 1GB, UK; m.i.cotterill@gmail.com (M.C.);; 6St Mary’s Hospital Imperial Healthcare Trust London, London W2 1NY, UK; 7Moriarty, Flynn & Barnes, Singapore 308900, Singapore; 8Faculty of Human Sciences, University of Tsukuba, Ibaraki 305-8577, Japan; 9School of Humanities, Kwansei Gakuin University, Hyogo 662-8501, Japan

**Keywords:** compassion-focused therapy, positive mental health, literature review

## Abstract

Background: Compassion-focused therapy (CFT) has been attracting attention in mental health practice and research. CFT is effective in reducing a variety of negative mental health symptoms. Positive mental health (PMH) focuses on an individual’s functioning, quality of life, and well-being, aiming to achieve fulfilment. A need for PMH has been increasingly recognised such as national policies incorporating recovery-oriented approaches. However, how effective CFT is for PMH outcomes remains to be investigated. This narrative review aimed to identify the literature that reports evidence on CFT used against PMH outcomes. Methods: Our research questions (RQs) were as follows: RQ1. What PMH outcomes are targeted in CFT intervention research? RQ2. Is CFT effective for PMH?” Medline, Embase, and PsycINFO were searched on the Ovid platform. All studies that mentioned “compassion focused therapy” and “compassion-focused therapy” were searched. Results: Sixteen RCTs were included published since 2012. Nine studies were from Europe, four from Asia, two from Northern America, and one from Australia and New Zealand. CFT was used for diverse PMH outcomes, and the effects were overall positive. Self-compassion and compassion were the most frequently evaluated outcomes. Conclusions: The mechanism of action for CFT on PMH needs to be evaluated. CFT can be used as part of personal recovery in mental health. More evidence from non-WEIRD countries including LMICs is needed.

## 1. Introduction

### 1.1. Compassion-Focused Therapy

Compassion-focused therapy (CFT) was developed by Paul Gilbert, aiming to reduce shame and self-criticism [1]. Shame and self-criticism are detrimental components in many mental health problems including depression. CFT integrates cognitive behavioural therapy with components from other sciences such as evolutionary psychology, social psychology, developmental psychology, Buddhist psychology, and neuroscience [2]. In working with clients, CFT involves psychoeducation focusing on compassion: compassion to others and to themselves (self-compassion). Compassion helps people regulate their emotions, leading to feelings of safeness, comfort and soothing. The key element of CFT is for the psychotherapist to have a compassionate mind to help clients experience compassion [3].

Despite its history being only about 20 years [4], numerous studies have reported the effects of CFT on mental health problems. A 2023 meta-analysis on clinical populations, based on 15 studies, found that CFT is especially effective for the improvement of outcomes such as self-compassion, self-criticism, depression, and eating disorders [5]. Likewise, an early systematic review published in 2015 (*n* = 14 studies) reported that CFT was effective for mental health problems, in particular for self-criticism [6]. Reviews about CFT for people with intellectual disabilities reported that CFT reduces shame and self-criticism and supports safeness, but the methodological reporting of the included studies was poor [7,8]. Evidence suggests that CFT is effective in reducing mental health problems.

CFT is understood to address shame and self-criticism by activating our soothing system that is related to compassion, contentment and safeness [3]. CFT’s emotional regulatory systems present that our emotions are categorised into three systems: threat system, drive system, and soothing system. The threat system is related to emotions such as fear anxiety or anger. The fight-or-flight response is one common response when the threat system is highly activated, protecting us from various dangers. The drive system is related to emotions such as excitement or pleasure, as this system is about pursuit, resource acquisition, and achievement. In CFT, mental distress is thought to arise from the overuse of these two systems [3]. When your threat system is foregrounded, you may feel anxious, fearful, or angry, then stressed. When your drive system is foregrounded, you may be temporarily excited about something you are pursuing, but if you do not obtain it, you feel ashamed and self-critical. On the other hand, the soothing system relates to contentment where we have no threats to protect ourselves from, and no goals that must be achieved. The soothing system can foster social safety, curb excessive resource-seeking and threat-oriented behaviour, and promote feelings of safeness and contentment, which are strongly linked with mental health [9]. Many CFT exercises aim to get in touch with the soothing system by developing a compassionate mind, hence a sense of safeness, contentment and calmness, which in turn can address shame and self-criticism.

### 1.2. Positive Mental Health

Positive mental health (PMH) aims to achieve fulfilment by promoting an individual’s functioning, quality of life, and well-being [10]. PMH is different from traditional mental health, which mainly focuses on the prevention and reduction of negative mental health symptoms including depression and anxiety [11]. PMH helps people cope with daily life stress, have a meaningful occupational life, and connect with their community [12]. Promoting PMH can prevent people from mental health problems, and facilitate recovery from mental health problems [13].

PMH can be categorised into two types. Hedonism attends to positive emotions and eliminates negative ones. Eudaimonism aims to achieve one’s potential and to live well [14]. Combining those two types, PMH is thought relevant to emotional, psychological, and social well-being. The six characteristics of PMH are as follows: (a) problem-solving and self-actualisation; (b) personal satisfaction; (c) autonomy; (d) interpersonal relationship skills; (e) self-control; and (f) prosocial attitude [15]. These are essential to living well. Research about PMH is needed.

Despite the high need for both CFT and PMH, evidence for CFT on PMH has not been investigated in depth. Theoretically, it makes sense to think that accessing our soothing system, facilitated by CFT, will improve PMH outcomes. Indeed, CFT on self-esteem was investigated in a meta-analysis (*n* = 10 studies) [16]. The effect size of CFT on self-esteem was medium. However, many other PMH outcomes remain to be explored.

### 1.3. Study Aims

This narrative review aimed to identify studies that report the effect of CFT on PMH. We chose this type of review to provide an overview of CFT on PMH, as it was a rather under-explored area [17,18]. Our findings would be especially helpful (a) to CFT practitioners and educators who aspire to apply CFT to PMH outcomes, and (b) to mental health policy-makers who need effective approaches for supporting PMH outcomes.

## 2. Materials and Methods

The design of this review is a narrative review. Our research questions (RQs) were as follows:
RQ1: What PMH outcomes are targeted in CFT intervention research? andRQ2: Is CFT effective for PMH?”

The inclusion criteria for articles were clarified using the PEO (Population, Exposure, and Outcomes) framework: (P) anybody, (E) CFT, and (O) PMH outcomes such as quality of life, well-being, fulfilment, and resilience. We retrieved these outcomes by reviewing systematic reviews on PMH [19,20]. We did not specify the population because, as noted above, we were aware that this area of research was thin. Additional filters were randomized controlled trials (RCTs) and being published in English in a peer-reviewed journal. Table 1 summarises the eligibility criteria.

Medline, Embase, and PsycINFO were searched from the Ovid platform as these databases were relevant to mental health and psychotherapy research. All studies that mentioned “compassion focused therapy” and “compassion-focused therapy” were searched using the multi-purpose function (i.e., “compassion focused therapy.mp.”). Six hundred and ninety-three studies were retrieved by JB, which was reviewed by YK: 152 on Medline, 206 on Embase, and 335 on PsycInfo. Duplicates (*n* = 17) were then removed. The remaining 676 articles were title/abstract-screened by JB, after concordance established from reviewing 10% of the 676 article (68 articles) with YK. Forty-eight articles were deemed possibly includable, so were full-text reviewed by JB again after 10% (*n* = 5) concordance established with YK. Finally, 16 were included, which were confirmed by AME, MC, and AK. PRISMA flowchart is presented below (Figure 1).

## 3. Results

Sixteen papers were included (Table 2).

### Targeted Outcomes

Diverse PMH outcomes were targeted: self-compassion, compassion, resilience, self-reassurance, mindfulness, life satisfaction, emotional regulation, well-being, psychological flexibility, meaning of life, coherence, breastfeeding satisfaction, quality of life, positive affect, gratitude, motivation, relaxation, empathy, and clinical improvement. Self-compassion was the most frequently targeted outcome, reported in 10 studies [21,26,28,35,37,40,42,45,47,48], followed by compassion in four studies [24,26,31,50].

Overall, improvements were reported for most of the PMH outcomes. No significant improvements were found in self-reassurance, emotion regulation psychological well-being (PWB) for the general population in France [26], psychological flexibility and breastfeeding satisfaction for mothers of infants [31], motivation for intervention, empathy, self-compassion, and relaxation for people with severe head injury [45].

To answer our RQs, CFT studies targeted various PMH outcomes. Self-compassion and compassion were more common than the other outcomes (RQ1). CFT’s effect on those outcomes were, in general, positive. Only 3 studies of 16 reported non-significant improvement in PMH outcomes. Effects on self-compassion and compassion were all positive, except for one study that evaluated self-compassion among people with severe head injury [45] (RQ2).

## 4. Discussion

This review evaluated CFT on PMH outcomes. Sixteen studies have been published since 2012. CFT was used for diverse PMH outcomes, and the effects were overall positive. Self-compassion and compassion were the most frequently evaluated outcomes.

Our findings can add new evidence to the CFT research landscape which predominantly focuses on its positive effect on a range of negative mental health outcomes [52]. Our findings demonstrate that CFT is not only helpful in reducing negative MH outcomes, but also is helpful in improving diverse PMH outcomes such as resilience, self-reassurance, mindfulness, life satisfaction, well-being, meaning of life, quality of life, gratitude, and motivation. One possible explanation for this is that CFT allows individuals to access their soothing system. The soothing system is one of the three emotional regulatory systems of CFT and is related to the activation of the body’s parasympathetic nervous system to induce a state of calmness, soothing, and safety [3]. Recent studies report facilitating calmness and soothing are not only effective for reducing negative symptoms but also effective for improving PMH outcomes [53,54,55]. This suggests that CFT can be used for nonclinical populations who want to augment their PMH resources [56,57]. However, it is also important to note the cases where CFT did not improve PMH outcomes significantly in populations such as the French general population [26]. Implementations of CFT in nonclinical fields may need to be further encouraged.

CFT also de-escalates the other two emotional regulatory systems: threat and drive systems [58]. The threat system is a mechanism designed to detect and respond to threats in the environment. The threat system relates to our sympathetic nervous system, which prepares us for fight, flight, or freeze responses when faced with perceived danger or threat. The drive system is associated with our desires, ambitions, and motivations. The emotions that the drive system can bring include excitement, enthusiasm, and determination. These emotions are conditional to whether rewards pursued are obtained or not, and when not obtained, we will experience shame and self-criticism. CFT calms these systems down, which in turn can help us focus on PMH [3]. Calming down the threat and drive systems leads to a reduction in negative symptoms such as shame and self-criticism, which result in augmented PMH outcomes [27]. These two approaches—directly supporting PMH outcomes and calming down negative emotions—may illustrate how CFT supports PMH outcomes. These mechanisms of action need to be empirically evaluated.

CFT’s effectiveness on PMH outcomes suggests its adaptation to personal recovery in mental health. Personal recovery in mental health is described as the subjective process of taking control of one’s life (including one’s symptoms), having optimism for the future and taking personal responsibility for one’s own recovery [59]. Many national policies such as ones in the United Kingdom, Australia, and Canada, highlight the importance of providing mental health interventions that are oriented towards personal recovery [60,61,62]. Growing support for recovery-oriented interventions has been recognised by the World Health Organization, locating personal recovery as the guiding vision of mental health systems internationally [63]. Compassion-based approaches can be used to facilitate personal recovery. For example, a randomised controlled trial identified that a self-compassionate writing intervention can improve self-rated self-compassion, proactive coping, mental health, and physical health [64]. Moreover, a systematic review identified that compassion-focused imagery intervention studies often report improvements in self-compassion and positive affect and reductions in self-criticism, shame, and paranoia for both clinical and non-clinical adult populations [65]. Outcomes such as improved coping, self-compassion, and reduced self-criticism are similar to the outcomes of recovery-oriented interventions [66]. Introducing compassion-based interventions, such as self-compassionate writing and compassionate imagery, to the public through recovery-oriented interventions may subsequently benefit personal recovery on a wider scale. For example, Recovery Colleges (RCs) are a globally used, recovery-oriented approach offering mental health education to the public [67,68]. RCs can introduce compassion-based techniques and/or psychoeducation informed by CFT to help improve the mental health of people, including those who do not access conventional services.

More international studies are needed in CFT. Out of the 16 included studies, only 4 were from a non-Western country, having taken place in Japan and the developing country of Iran. This reflects the current landscape of mental health research where populations within Western, Educated, Industrialised, Rich, and Democratic (WEIRD) countries are overrepresented in social and behavioural science research, despite accounting for 12% of the global population [69,70]. The overrepresentation of WEIRD data in social and behavioural research contributes to a culture of “fundamental attribution error” where there is a disregard for behaviours being intertwined within one’s cultural context, in favour of attributing behaviour to individual personal traits [71]. This error relates to our tendency to interpret other people’s behaviour based on their personality or internal qualities while ignoring the influence of situational factors [72]. Self-compassion is governed by various cultural practices and group norms [73]. For example, in some Asian cultures, high levels of self-criticism and low levels of compassion have been attributed to notions of self-improvement being a function of shame [74]. In some low- and middle-income countries (LMICs), in order to sustain their daily living, contentment may not be valued in the same way as it is in high-income countries. CFT evidence from non-WEIRD countries needs to be collected.

Especially CFT evidence from LMICs is valuable. Eighty percent of the global population who have mental disorders live in LMICs and researchers expect depression (which often encompasses shame and self-criticism) to be the third leading cause of mental distress in LMICs by 2030 [75]. While interventions to reduce negative symptoms in LMICs have been proposed [76], how to augment PMH in people in LMICs is under-researched. Simple, low-cost, and self-help CFT exercises such as letter writing, can be recommended to support PMH of people in LMICs. Digital tools can be used to facilitate this process and have been shown to be useful in LMICs [77].

Limitations of this review need to be noted. First, this was not a systematic review therefore only three databases were searched, and a quality assessment for the included studies was not carried out. Our aim was to provide an overview of CFT on PMH, so a narrative review was chosen as the design instead of targeting more specific outcomes (e.g., a meta-analysis published during this review that evaluated the CFT effects on compassion [78]). Second, grey literature and studies in non-English languages were not included. The authors were not aware of RCTs in the effect of CFT on PMH outcomes in any other languages. Third, three studies reported [26,31,45] non-significant results. As those studies were heterogeneous, we were not able to further discuss them; however, future research can explore outcomes, populations and contexts where CFT does not work. Similarly, weaknesses of CFT such as unidentified mechanisms of actions or components [79] were not considered in this review. Finally, all authors were researchers and/or practitioners in mental health. We might have missed an uncommon influential factor in mental health research.

## 5. Conclusions

This review evaluated the effect of CFT on PMH outcomes. Diverse PMH outcomes were targeted in CFT studies, where self-compassion and compassion were the most common outcomes. Overall positive effects were reported. CFT’s mechanism of accessing our soothing mind may help explain its effect on PMH. CFT application to recovery approaches and implementation in various settings need to be developed in future research. Our findings will help researchers and practitioners to understand the current status of CFT on PMH, and to plan future research and applications.

## Figures and Tables

**Figure 1 behavsci-14-00643-f001:**
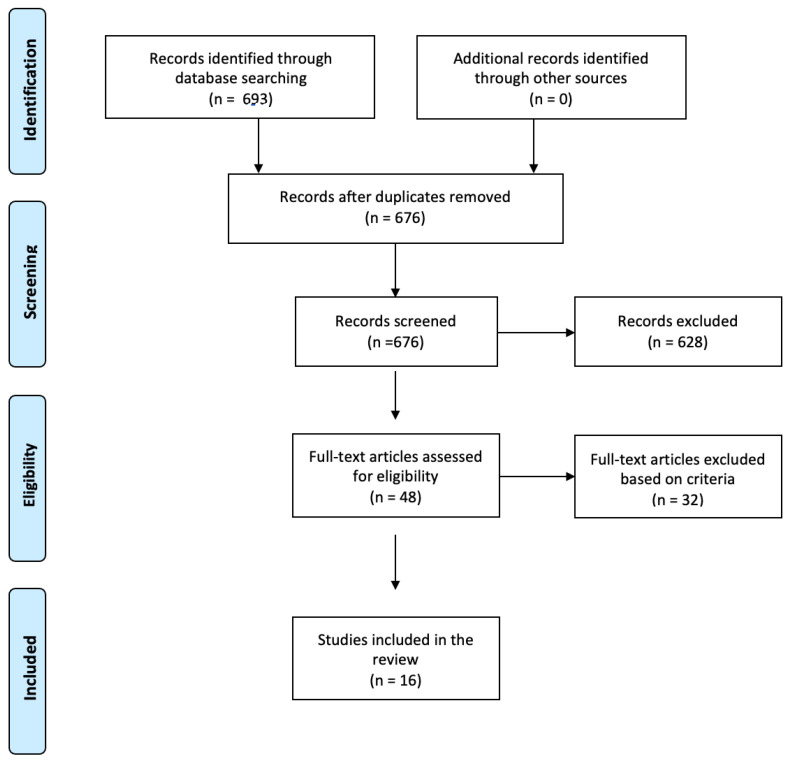
PRISMA flowchart.

**Table 1 behavsci-14-00643-t001:** Eligibility criteria.

	Include	Exclude
Population	No restriction	NA
Exposure	Compassion-focused therapy	Other therapies, including other forms of compassion-based interventions (e.g., acceptance and commitment therapy)
Outcomes	Positive mental health outcomes such as quality of life, well-being, fulfilment, and resilience.	Other outcomes such as depression, anxiety, addiction
Design	RCTs	Other types such as pre-post, qualitative studies, case notes, reviews
Language	English	Other languages
Types	Peer-reviewed articles	Other types of publications including theses, dissertations, magazine articles, books
Time period	No restriction	
RQ1: What PMH outcomes are targeted in CFT intervention research? RQ2: Is CFT effective for PMH?		

**Table 2 behavsci-14-00643-t002:** Data abstraction table.

	Author, Year, Country	Intervention Name and Reference	Control	Session Number and Duration	Treatment Protocol Available?	Session Topics	Facilitator(s)	Facilitator Training	Outcomes	Effective?
1	Kopland et al. (2023), Norway [21]	CFT or CBT [22,23]	Active	13-week treatment. 2–3 group sessions per week per treatment group. Three 45 min individual sessions per week	None	Weekly outcome and process questionnaires each Monday	9 trained clinical psychologists and 1 therapist in the CBT condition 6 clinical psychologists and 3 psychiatrists in the CFT-E group	All therapists had formal clinical training in CFT-E or CBT.	**SCS**; Eating disorder symptoms (EDE-Q)	Yes: both outcomes. No difference between CFT and CBT.
2	Asano et al. (2022), Japan [24]	Group CFT [1]	Active	12-week CFT sessions (90 min per session)	None	Psychoeducation about compassion, depression, mindfulness for sound, three-circle model, and the mechanisms of emotion and motivation, soothing rhythm breathing Compassion exercises including imagery, compassionate self, compassionate memory, the use of compassionate images, CFT case formulation, replacing safety behaviours, compassionate letter writing exercise	Trained Clinical Psychologist with a PhD in Psychology and an Industrial Counsellor conducted the sessions.	Clinical Psychologist attended a three-day CFT workshop. Both facilitators had completed a CBT training course	Depressive symptoms (BDI and GRID-HAMD); Fears of compassion (FCS); **CEAS**	Yes: all outcomes
3	Jalayer et al. (2022), Iran [25]	CFT group therapy	Active	10 sessions	No	Accepting mistakes and self-forgiveness, understanding, compassion and appreciation. Understanding pleasant feelings and conflicting emotions.	Not recorded.	Not recorded.	Emotional schemas (LESQ); **Resilience (CDRQ)**	Yes: both outcomes.
4	Leboeuf et al. (2022), France [26]	Compassionate Mind Training [27]	Inactive	28 daily exercises for 4 weeks lasting less than 15 min	None	Psychoeducation on CMT, e.g., thoughts, feelings and emotions linked to compassion and fears Exercises, e.g., smile, soothing breathing rhythm, place of serenity, behavioural activations of positive relationships, and behavioural activation of receiving compassion	First/corresponding author and fifth author of the study	Not recorded	**Compassion (CEAS)**; Anxiety (STAI); Stress (PSS); Self-criticism and **self-reassurance (FSCRS)**; **Mindfulness (FFMQ)**; **Self-compassion (SCS)**; Depression (BDI); **Life satisfaction** (SWLS); **Emotion regulation (PEC);** **Psychological well-being (PWB)**	Yes: compassion, self-compassion and mindfulness. No: the other outcomes
5	Carvalho et al. (2021), Portugal [28]	Acceptance and Commitment Therapy [29] and CFT [1]	Active	4 weekly sessions delivered online over 4-weeks. 20 min and included meditative practices (10–20 min)	None	The ACT treatment session topics included introduction to mindfulness and mindfulness of breathing. The CFT treatment included sessions related to compassion and loving, kindness meditation, self-compassion, compassionate self, fears of compassion and compassion meditation	Not recorded	Not recorded	Anxiety and depression (HADS); ACT assessment (CompACT); Chronic illness related shame (CISS); Cognitive fusion (CFQ-CI); **SCS**	Yes: all outcomes. No significant difference between CFT and ACT
6	Daneshvar et al. (2020), Iran [30]	Group CFT [1]	Inactive	8 sessions, 2 hrs per session	None	Sessions included defining trauma, training empathy, sympathy, forgiveness, acceptance towards events, to develop valuable and transcendental feelings, commitment toward self, and training and reviewing skills	Not recorded	Not recorded	Experiential avoidance and **psychological flexibility** (AAQ); **Meaning of life (MLQ); Coherence (SoC);**	Yes: experiential avoidance, psychological flexibility, and meaning of life No: Coherence
7	Lennard et al. (2020), Australia and New Zealand [31]	Brief self-compassion intervention [32]	Inactive (waitlist)	8 weeks	None	Self-compassion, fears of compassion, psychological flexibility, depression, anxiety, stress, symptoms of posttraumatic stress, and infant feeding experiences	Not recorded	Not recorded	**CEAS**; FCS; **AAQ**; Depression, anxiety and stress (DASS); PTSD (IES-R); **Breastfeeding satisfaction (MBES).**	Yes: compassion, depression, posttraumatic stress No: fears of compassion, psychological flexibility, anxiety, stress, and breastfeeding satisfaction.
8	Sommers-Spijkerman et al. (2019), Netherlands [33]	Compassion as key to happiness’ self-help book [34]	Inactive (waitlist)	7 lessons over 7–9 weeks (self-administered)		Psycho-education on compassion, self-reflective and experiential exercises.	All participants were randomly assigned to one of five counsellors. Counsellors were the first and second author, one graduated psychologist and two Master students Psychology. All five counsellors were trained and supervised by two experienced health care psychologists (i.e., fourth and fifth author).	1st author developed the protocol and psychology students were trained and supervised 2nd and 3rd authors (experienced healthcare psychologist).	Total frequency of the respective compassionate attribute/skill; Total frequency of the attribute/ skill	Yes: five compassionate attributes (i.e., care for well-being, sensitivity, empathy, distress tolerance, and common humanity) and four compassionate skills (i.e., compassionate attention, reasoning, behaviour, and feeling).
9	Gharraee et al. (2018), Iran [35]	CFT [36]	Inactive	12 weekly 1hr individual sessions	Boersma et al. (2015) [36]	Psychoeducation on shyness, brain evolution and threat detection systems, emotional regulation systems, shame and self-criticism and barriers to compassion. ‘homework’ included imagery and compassion exercises.	1st author (Trainee Clinical Psychologist)	Not recorded	Major psychiatric disorders (Structured Clinical Interview for DSM-IV Axis I Disorders, Clinician Ver); Social Anxiety (LSAS); **WHOQ-QoL; AAQ; SCS**; Mindfulness (MAAS); Self-criticism (LSCS)	Yes, for all variables.
10	Sommers-Spijkerman et al. (2018), Netherlands [37]	CFT [1]	Inactive (waitlist)	1 lesson per week for a total of 9 weeks	Internet based mindfulness treatment [38,39]	Exercises included mindful breathing, keeping a diary of self-critical thoughts, visualising one’s ideal compassionate self and expressing compassion for someone else in a letter	Not recorded	Not recorded	**Well-being (MHC-SF)**; HADS; PSS; **FSCRS; SCS-SF**	Yes: all outcomes, in particular improvement in well-being
11	Sommers-Splijerman et al. (2018), Netherlands [40]	Compassion as key to happiness’ self-help book [34]	Inactive (waitlist)	7 lessons over 7–9 weeks (self-administered)	Hulsbergen & Bohlmeijer, 2015 [34] and Sommers-Spijerman et al. 2018 [41]	Psycho-education on compassion, self-reflective and experiential exercises to cultivate compassionate attributes and skills.	Self-administered with e-mail guidance from first author, two psychologist graduates and two psychology master’s students.	First author developed the protocol and psychology students were trained and supervised second and third authors (experience healthcare psychologist).	**MHC-SF**; HADS; PSS; **FSCRS; SCS-SF**; **PANAS;** **Gratitude (GQ6);** **Measure the satisfaction with the information in the self-help book**	Yes: all outcomes
12	Ascone et al. (2017), Germany [42]	CFI [1]	Active	1 short session on CFI	Gilbert, 2010 [3]	Negative emotion induction. Creating images that elicited feelings of warmth and compassion to themselves.	First and third authors	Facilitators practiced the exercise script until they were able to deliver it in a calm and warm tone.	**FSCRS; SCS;** Fear and anger [43]; Skin Conductance Levels; (Q sensor 2.0); Paranoia (Paranoia Checklist); Appraisal and Perceived Subjective Benefit of Imagery Intervention; Affective Valence and Arousal [44]	Yes: self-relating and positive affect (self-reassurance, self-compassion and happiness) No: negative self-relating (self-criticism, negative affective states (i.e., shame, depression, anxiety and anger)); psychophysiological arousal (skin conductance); paranoia
13	Campbell et al. (2017), UK [45]	CF imagery (CFI) [46]	Active	20 min preparatory video 50 min CFI or relaxation imagery intervention	Supplementary material provided in [45]	Participants were given a video task and one imagery intervention	Not recorded	Not recorded	FCS; **Motivation for intervention (MIS);** Negative affect only (PANAS); STAI; **Empathy (EQ)**; **SCS-SF; Relaxation (RS);** Heart rate variability (HRV)	No: all outcomes
14	Kelly et al. (2017), Canada [47]	Group CFT [1]	Active (TAU)	12 × 90 min group sessions over 12 weeks.	Kelly & Leybman 2012, unpublished, see [47]	Psycho-education of how the brain has evolved, affect regulation, importance of compassion and self-compassion. How compassion can alleviate shame and self-criticism. Barriers to compassion.	Psychologist with CFT training and a master’s level therapist.	Facilitators had regular supervision from the first author who has had extensive CFT training and supervision from Gilbert.	Credibility and expectancy (CEQ); EDE-Q; **SCS**; FCS; Shame (ESS).	Yes: shame, self-compassion, fears of compassion and self-compassion, and eating disorder pathology
15	Kelly & Carter (2015), Canada [48]	CFT-based self-help intervention for binge eating disorder [22,23,49]	Active and Inactive (Waitlist)	12 weeks		CBT based help book for binge eating. Psychoeducation of self-compassion and binge eating using imagery, self-talk and writing	Psychology researcher	Not recorded	EDE-Q; ESS; **SCS-SF**	Yes: all outcomes
16	Braehler et al. (2012), UK [50]	CFT group therapy for recovery after psychosis [51]	Active (TAU)	16 × 2hr group session each week over 4–5 months.		Psychoeducation on psychosis and compassionate motivation Developing shared meaning of compassion. Developing compassion within self. Compassionate skills training.	Each group was led by two psychologist and five trial therapists.	All had experience of facilitating psychological therapy for psychosis. 4/5 received training at the 3-day workshop on CFT led by Gilbert. Fortnightly peer group supervision and frequent consultation with Gilbert.	Avoidance and **compassion** (NRSS); **Improvement** and exacerbation (CGI-I); BDI; **PANAS**; Fear of recurrence (FoRSe); Illness beliefs (PBIQ-R)	Yes: all outcomes

**Positive outcomes are bold.** CEAS = Compassionate Engagement and Action Scales; STAI = Spielberger State-Trait Anxiety Inventory; PSS = Perceived Stress Scale; FSCRS = Forms of Self-Criticising/Attacking & Self-Reassuring Scale; FFMQ = Five Facet Mindfulness Questionnaire; SCS = Self-Compassion Scale; SCS-SF = Self-Compassion Scale-Short Form; BDI= Beck Depression Inventory; SWLS = Satisfaction With Life Scale; PEC = Profile of Emotional Competence; PWB = Psychological Well-being Scale; HADS = Hospital Anxiety and Depression Scale; CompACT = Comprehensive assessment of Acceptance and Commitment Therapy processes; CISS = Chronic Illness-related Shame Scale; CFQ-CI = Cognitive Fusion Questionnaire-Chronic Illness; MHC-SF = Mental Health Continuum-Short Form; GRID-HAMD = GRID-Hamilton Depression Rating Scale; FCS = Fears of Compassion Scale; EDE-Q = Eating Disorder Examination Questionnaire; AAQ = Acceptance and Action Questionnaire; SoC = Sense of Coherence Scale; MLQ = Meaning in Life Questionnaire; NRSS = Narrative Recovery Style Scale; LESQ = Leahy Emotional Schemas Questionnaire; CDRQ = Conor and Davidson’s Resilience Questionnaires; LSAS= Liebowitz Social Anxiety Scale; WHOQ-QoL = World Health Organization Questionnaire of the Quality of Life); MAAS = Mindful Attention Awareness Scale; LSCS = Level of Self-Criticism Scale; CEQ = Credibility and Expectancy Questionnaire; DASS = Depression Anxiety Stress Scale; IES-R = Impact of Event Scale–Revised; MBES = Maternal Breastfeeding Evaluation Scale; MIS = Motivation for Intervention Scale; PANAS = Positive and Negative Affect Schedule; EQ = Empathy Quotient; RS = Relaxation Scale; GQ6 = Gratitude Questionnaire-6-Item Form; ESS = Experiential Shame Scale; CGI-I = Clinical Global Impression–Improvement Scale; FoRSe = Fear of Recurrence Scale; PBIQ-R = Personal Beliefs about Illness Questionnaire-Revised. TAU = Treatment as usual. The publication years ranged from 2012 [50] to 2023 [21]. Nine studies were from Europe [21,26,28,33,37,40,42,45,50], four from Asia [24,25,30,35], two from Northern America [47,48], and one from Australia and New Zealand [31]. Three studies were from the Netherlands [33,37,40] and Iran [25,30,35], respectively. Gilbert’s book [1] was the most used intervention guidance, informed the interventions in six studies [24,28,30,37,42,47].

## Data Availability

Not applicable.

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
