# Peer review of "A Narrative Review of Compassion Focused Therapy on Positive Mental Health Outcomes"

_behavsci, 2024, doi:10.3390/bs14080643_

Round 1
Reviewer 1 Report (Previous Reviewer 1)
Comments and Suggestions for Authors
The authors have addressed all the requested revisions comprehensively, significantly enhancing the clarity and rigor of the paper. I don't have any suggestions.
Author Response
Comment 1: The authors have addressed all the requested revisions comprehensively, significantly enhancing the clarity and rigor of the paper. I don't have any suggestions.
Response 1: Thank you for your feedback.
Reviewer 2 Report (New Reviewer)
Comments and Suggestions for Authors
Thank you for this work. A few suggestions:
Given these were all RCTs, it would be nice to see what the comparison groups are in the table. Are they active controls?
One small area of improvement would be to move up your rationale for doing a narrative review to the aims section of the paper and include a reference.
Finally, it seemed as if the discussion was focused more on mechanisms of action than the PMH outcomes that were covered in the paper. I would have loved to hear more on the specific PMH outcomes examined and why and for whom they are important.
Author Response
Comment 1: Given these were all RCTs, it would be nice to see what the comparison groups are in the table. Are they active controls?
Response 1: In line with your comment, controls are now added.
Comment 2: One small area of improvement would be to move up your rationale for doing a narrative review to the aims section of the paper and include a reference.
Response 2: In line with your comment, now our rationale for narrative review has been integrated into “study aims”.
Comment 3: Finally, it seemed as if the discussion was focused more on mechanisms of action than the PMH outcomes that were covered in the paper. I would have loved to hear more on the specific PMH outcomes examined and why and for whom they are important.
Response 3: Thank you. Now more discussion on outcomes and impact are included in the discussion.
This manuscript is a resubmission of an earlier submission. The following is a list of the peer review reports and author responses from that submission.
Round 1
Reviewer 1 Report
Comments and Suggestions for Authors
Introduction and Present Study
The introduction briefly summarize the CFT theoretical foundation and its comprehensive integration in all psychological domain is quite effective.
However, could you try to explicitly inform the reader how exactly CFT addresses mental disorders and directly reduces mental health problems to the reduction of shame and self-criticism? While the statement clearly refers to the previous 2023 and 2015 meta-analysis studies confirming the effectiveness of CFT on various outcomes, could you provide a summary of the studies explaining how your study adds something new to this evidence?
Study objectives. The statement about thematic synthesis of evidence is quite general. Could you elaborate on the methodology you plan to use for such a synthesis? What is the process of a thematic synthesis and what makes it more valuable than other reviews?
Methods
The manuscript refers to the review as a "general literature review". This term could be interpreted in many ways. Could you be a little more specific about the nature of the review? Should it be systematic, narrative, scoping or some other kind of review?
Would it also be possible to follow established guidelines more closely if the review is systematic? For example, to ensure methodological clarity and rigour, Moher et al suggest following the PRISMA guidelines.
The authors report the number of studies that were retrieved, the number of studies that were potentially included after an initial screening, and the final number of studies that were included in their review. However, they do not provide detailed information about their screening and selection of studies.
Could you explain the steps involved in screening and selection? In addition, it is recommended that authors of systematic reviews use a PRISMA flowchart that describes the process by which the studies to be reviewed were screened and assessed for eligibility or inclusion and exclusion. Could you include such a diagram in your manuscript to provide more information to your readers?
There is no mention in the text of data extraction and quality assessment of eligible studies. Could you provide more information about the data extraction process for the studies included in your review? Did you use any tools or checklists to assess the methodological quality of the studies? For example, did you use the Critical Appraisal Skills Programme checklist or any other standardised tool to assess study quality? It is important to understand how you approached these components.
Finally, while you mention thematic synthesis, there is no detail on how you carried out this synthesis. Can you provide more information on how you identified and synthesised the themes across the included studies?
How was consistency between reviewers ensured in the screening process and in the full-text review? Can you describe any specific strategies or techniques used to ensure that the reviewers' independent assessments and decisions were congruent, given that the selection of studies should be as free from bias as possible to increase reliability?
Finally, did the review include non-English language publications? If not, what was the justification for such a language barrier? As it would help to identify non-English language studies and assess the potential bias due to language barriers in the review conclusion.
Results
Considering the systematic review's structure, could you provide a more detailed synthesis of the results pertaining to each specific research question outlined in your study? How does the efficacy of CFT vary according to the different PMH outcomes addressed?
In light of the general positive impact of CFT on PMH outcomes reported, could you offer a nuanced discussion on any variations or patterns in the efficacy of CFT? This might include considerations of study design, population demographics, or intervention duration that could influence the observed outcomes
Given the importance of reproducibility and transparency in reviews, it would be highly valuable if supplementary materials could be provided that detail the specific search strings used by the authors for each database consulted in the study. Including such information would not only enhance the review's utility for future research but also allow for a more thorough assessment of the comprehensiveness of the search strategy employed. Could you please consider adding these details as supplementary materials?
To further strengthen the breadth and depth of your systematic review, particularly given its thematic focus, I recommend considering the inclusion of additional databases beyond the psychological-clinical ones currently listed (Web of Science, PubMed).
Expanding your search to encompass databases that cover broader interdisciplinary fields, such as Scopus, EBSCOhost, and perhaps even subject-specific databases relevant to CFT and PMH outcomes, could provide a more comprehensive view of the literature. Incorporating a wider array of databases might uncover additional studies of relevance, thereby enriching the review's findings and conclusions. Could you explore the feasibility of extending your database search in this manner?
Discussion
I would like to suggest the possibility of refining the methodology of this review to align more closely with that of a systematic review. Such a modification could not only enhance the methodological rigor of your work but also increase the transparency and replicability of the study. Systematic reviews, by following well-defined protocols such as PRISMA guidelines, provide a structure for exhaustive literature search, clearly defined inclusion and exclusion criteria, systematic assessment of the quality of included studies, and, where appropriate, the capability to perform meta-analysis. These elements significantly contribute to the reliability and validity of the conclusions drawn.
I recognise that transforming this review into a systematic review would involve considerable additional effort, including possibly extending the bibliographic search to include studies published in languages other than English and undertaking a systematic quality assessment of the included studies.
However, I believe that the benefits in terms of improved quality of evidence and practical applicability justify such an effort. In addition, adapting to a systematic review format could provide an opportunity to explore variations in study results based on factors such as cultural and geographical context, further enriching our understanding of the effectiveness of CFT.
Reviewer 2 Report
Comments and Suggestions for Authors
Thank you for the opportunity to review. This manuscript surveys the current state of CFT intervention research on PMH outcomes. I think it's condensed to the point, but there are a few things I'd like to confirm.
#1. What is the rationale and standard program of CFT? The rationale for CFT seems to be divided into an introduction section and a discussion section.
#2. How did the authors define the inclusion criteria for PMH outcomes?
#3. Regarding lines 158 to 161 of the discussion section. The authors mentioned that “Recent studies report facilitating calmness and soothing are not only effective for reducing negative symptoms but also effective for improving PMH outcomes [51-53]. This suggests that CFT can be used for nonclinical populations who want to augment their PMH resources [54,55].” What did the authors think about the mechanism of action of CFT? Does PMH improve by reducing negative symptoms, or does negative symptoms reduce by cultivating PMH?
